# Validation of Fitbit Charge 4 for assessing sleep in Chinese patients with chronic insomnia: A comparison against polysomnography and actigraphy

**Xiaofang Dong**[1☯], **Sen Yang**[2☯], **Yuanli Guo**[1☯], **Peihua Lv**[1☯], **Min Wang**[1☯], **Yusheng Li**[1☯]*

**1** Neurology Department, The First Affiliated Hospital of Zhengzhou University, Zhengzhou, Henan Province, China, **2** Orthopedics Department, The Seventh Hospital of Zhengzhou, Zhengzhou, Henan Province, China

☯ These authors contributed equally to this work.
* fccliyusheng@zzu.edu.cn

**Data Availability Statement:** Data relevant to this study are available from Science Data Bank at: DOI:10.57760/sciencedb.j00001.00453 (http://doi.org/10.57760/sciencedb.j00001.00453).

## Abstract

Our research aims to assess the performance of a new generation of consumer activity trackers (Fitbit Charge 4[TM]: FBC) to measure sleep variables and sleep stage classifications in patients with chronic insomnia, compared to polysomnography (PSG) and a widely used actigraph (Actiwatch Spectrum Pro: AWS). We recruited 37 participants, all diagnosed with chronic insomnia disorder, for one night of sleep monitoring in a sleep laboratory using PSG, AWS, and FBC. Epoch-by-epoch analysis along with Bland–Altman plots was used to evaluate FBC and AWS against PSG for sleep-wake detection and sleep variables: total sleep time (TST), sleep efficiency (SE), waking after sleep onset (WASO), and sleep onset latency (SOL). FBC sleep stage classification of light sleep (LS), deep sleep (DS), and rapid eye movement (REM) was also compared to that of PSG. When compared with PSG, FBC notably underestimated DS (-41.4, $p < 0.0001$) and SE (-4.9%, $p = 0.0016$), while remarkably overestimating LS (37.7, $p = 0.0012$). However, the TST, WASO, and SOL assessed by FBC presented no significant difference from that assessed by PSG. Compared with PSG, AWS and FBC showed great accuracy (86.9% vs. 86.5%) and sensitivity (detecting sleep; 92.6% vs. 89.9%), but comparatively poor specificity (detecting wake; 35.7% vs. 62.2%). Both devices showed better accuracy in assessing sleep than wakefulness, with the same sensitivity but statistically different specificity. FBC supplied equivalent parameters estimation as AWS in detecting sleep variables except for SE. This research shows that FBC cannot replace PSG thoroughly in the quantification of sleep variables and classification of sleep stages in Chinese patients with chronic insomnia; however, the user-friendly and low-cost wearables do show some comparable functions. Whether FBC can serve as a substitute for actigraphy and PSG in patients with chronic insomnia needs further investigation.

**Funding:** This study was supported by the Key Scientific Research Project of Henan Province (grant numbers No. 14 HWYX2021057) and the Key Scientific Research Project Plan of Colleges and Universities in Henan Province(No. 22A320023), awarded to XD. The funders had no role in study design, data collection and analysis, decision to publish, or preparation of the manuscript.

**Competing interests:** The authors have declared that no competing interests exist.

## Introduction

Adults (over 18 years) should sleep at least 7 hours in the evening to maintain the best health status [1]. Epidemiological studies' results demonstrate that lack of sleep is linked to multiple diseases, risk of accidents, and increased mortality [2–5]. Moreover, sleep disorders have become one of the most important social issues in the United States [6], with 35% of adults sleeping less than 7 hours every evening, 30% sleeping less than 6 hours, and more than 40% experiencing lack of sleep due to certain types of jobs [7]. Chronic insomnia is the most commonly reported sleep disorder and more than 15% of Chinese adults have difficulty falling asleep and maintaining sleep [8]. Despite the association between insufficient sleep and adverse health outcomes, most medical workers are unable to successfully identify sleep problems. It is estimated that unrecognized sleep problems are more common and harmful than recognized sleep problems [9].

Traditional methods for measuring sleep have some disadvantages. Polysomnography (PSG), for example, is considered the "gold standard" of sleep measurement owing to its objectivity and accuracy in measuring sleep variables [10]. However, being an expensive technique, PSG can only be carried out in an unfamiliar sleep environment, such as a sleep center, and requires medical assistance for its application and the interpretation of its results. This also means that it can typically only be implemented for one or two nights [11]. Furthermore, although sleep diaries are inexpensive and can be conducted outside the sleep center, they still have problems such as subjectivity and recall bias [11]. Actigraphy, the measurement of sleep and wakefulness using accelerometer technology according to activity level, can offer objective sleep variables for up to 60 days [12, 13] and support the diagnosis and monitoring of sleep problems [14]. Validation studies in infants, children, and adolescents have shown that while actigraphy is sensitive in assessing actual sleep, it over- or underestimates waking after sleep onset (WASO), thus providing a poor estimate of sleep disruption when compared to PSG [15–17]. Even though actigraphy is less expensive and more reflective of natural sleep patterns than PSG, it still needs professional researchers to extract and analyze data from a unique software [18].

Although wearables have been in the consumer market for just over ten years, they have become almost ubiquitous in modern society due to scientific and technological advancements. According to data from the International Data Corporation (IDC), 172.2 million units of wearable devices were shipped globally in 2018, with an increase of 49.2% year-on-year. It is expected that shipments will increase to 27.9 billion units in 2023 [19]. In view of their wide range of functions, it is crucial to determine wearables' most relevant indicators in the science of sleep and circadian rhythms. The rise of consumer wearables has offered another low-cost, good-looking, and user-friendly choice to assess sleep-wake patterns, which would be helpful considering clinical and research needs [19–21]. In addition to being low cost, these devices generally utilize consumer-oriented cloud computing platforms and/or mobile technologies that allow consumers to collect and retrieve data continuously for a certain period. Although there has been little study measuring sleep parameters by wearables in participants with sleep disorders, the enormous potential of these devices is acquiring more and more recognition among health and clinical studies. Moreover, these wearable devices have been used as auxiliary tools to diagnose and treat sleep disorders [22].

As one of the most promising wearables, the accurateness of Fitbit Charge 4™ has not been proved among Chinese participants with chronic insomnia. Previous validation studies compared various types of Fitbit devices, including Fitbit Classic [23], Fitbit Flex [24], Fitbit Ultra [18], Fitbit Charge 2 [21], and Fitbit Charge 3 [25] with PSG. The outcomes of these studies varied according to the model of commercial activity trackers, the demographic and clinical

state of the participants, study design, and data algorithm. It is thus necessary to implement utility research on clinical populations with sleep problems [23]. Although the majority of the published research was conducted on young volunteers, it appears that none of them focused on Chinese patients with chronic insomnia.

Therefore, the purpose of our research was to: (a) detect differences in sleep-wake detection and sleep variables (including total sleep time, TST; sleep efficiency, SE; sleep onset latency, SOL; and WASO) between the Actiwatch Spectrum Pro (AWS) and the Fitbit Charge 4$^{TM}$ (FBC) against PSG; and (b) examine whether similarities in sleep staging exist between FBC and PSG.

## Materials and methods

### Design and sample

Our research (registration ID: ChiCTR2100051429) was certified by the Research Ethics Committee of the First Affiliated Hospital of Zhengzhou University (approval number: 2021-KY-0876-001). All participants signed written informed consent forms before the sleep monitoring process in the sleep laboratory. The Standards for the Reporting of Diagnostic Accuracy (STARD) guidelines were followed where applicable [26].

Participants were recruited between October 2021 and February 2022 and included in the sample if they were: (a) between the ages of 18 and 60 years; (b) diagnosed with chronic insomnia, according to the DSM-5 criteria, for at least 3 months; (c) assessed as experiencing sleep disturbance according to the Pittsburgh Sleep Quality Index (PSQI), with a score above 8; and (d) receiving no treatment for the disorder during the preceding half month. Participants were excluded if they: (a) had suffered from mental illness; (b) had unhealed sleep disorders except for chronic insomnia, such as obstructive sleep apnea (OSA) and rapid eye movement sleep behavior disorder (RBD); (c) had other kinds of diseases (such as Parkinson's disease or dementia), which could impact their capacity to understand and deal with information; (d) worked night shift in the preceding half year; and (e) were gestating or lactating women.

### Clinical questionnaires

All the participants took the Chinese version of PSQI, the Insomnia Severity Index (ISI), and a self-designed questionnaire including general demographic information and sleep habits, such as age, gender, height, weight, time to sleep and wake up, and history of diseases [27].

### PSG, FBC, and AWS

**PSG measures.**   The participants were requested to arrive at the sleep center between 18:00 and 19:00 for their sleep monitoring. Standard overnight PSG recordings were carried out in the sleep laboratory with the computerized sleep recorder EMBLA S4500 (Embla System, Kanata, Ontario, Canada). An electroencephalography (EEG; F3/F4, C3/C4, O1/O2), electrooculogram (EOG-L, EOG-R), submental electromyogram (EMG), leg electromyogram (EMG-L, EMG-R), and left and right electrocardiograms (ECG-L, ECG-R) were obtained and recorded. Nasal pressure, abdominal breathing movements, and blood oxygen saturation were recorded to identify respiratory events. Sleep periods were set according to the times the lights were turned on and off.

**Actigraphy.**   Participants wore an Actiwatch Spectrum Pro (Philips Respironics Inc., Pittsburgh, Pennsylvania, United States) on their non-dominant hand for a research-grade actigraphy. Data for AWS were collected in 30-second epochs using the medium threshold

(value = 40), with five-minute immobility time for sleep onset/offset. This setup is proved to generate the best outcomes when compared with PSG [28].

**Fitbit Charge 4.** Each participant wore a consumer-grade activity tracker, the Fitbit Charge 4TM (Fitbit, Inc.; San Francisco, California, United States), on their non-dominant hand while the PSG was being performed. This device tracks motion, heart rate (HR), and heart rate variability (HRV) via accelerometers and optical plethysmography in 30-second epochs. Furthermore, the "normal" setting was chosen as the default setting to meet most consumers' preferences.

**Data collection.** The times at which the lights were turned on and off for PSG were considered the beginning and ending points for AWS and FBC [18]. Sleep stages were automatically scored in 30-second epochs using RemLogic 3.4.1 (Embla Systems, Kanata, Ontario, Canada) and visually reviewed by a trained technician (intra-rater reliability, 94.8%) following the criteria set by the AASM Scoring Manual Version 2.2 to ensure accuracy of the staging [29]. PSG registered five stages of sleep: awake, stage N1, N2, N3, and rapid eye movement (REM) sleep. Owing to the difficulty in distinguishing between the N1 and N2 sleep stages for Fitbit, a summation of N1 and N2 sleep stages in PSG was categorized as "light sleep" and N3 as "deep sleep" to compare with FBC. Data from AWS were processed using Actiware version 6.6.7 (Philips Respironics Inc., Pittsburgh, Pennsylvania, United States). Automatically analyzed data were refined according to lights off and on times from PSG. The FBC sleep information was extracted from the Fitbit web interface once the device synchronized with the interface via the Bluetooth capabilities of a USB-connected dongle. As soon as data synchronization was complete, the start and end time of the sleep cycles were manually revised to be consistent with that of PSG. FBC classifies epochs into awake or one of three sleep stages: light sleep (LS), deep sleep (DS), or REM sleep. Sleep stage was manually collected from summary figures on the web interface.

The sleep variables were TST (min), SE (%), SOL (min), WASO (min), stage N1 (min), stage N2 (min), stage N3 (min), and REM (min). In order to compare with PSG stage classifications, the sum of N1 and N2 assessed by FBC was assumed to represent LS (min) and N3, DS (min). All the authors earnestly protected the participants' privacy and none of the authors had access to information that could identify individual participants during or after data collection.

**Data analysis.** All statistical analyses were conducted using SPSS version 22.0 software (IBM Corp., Armonk, New York, United States) and MedCalc version 12.6 (MedCalc Software Ltd, Ostend, Belgium). Statistical significance was established at $p < 0.05$, with two-sided tests for every sleep variable. We calculated the mean and standard deviation(SD) of each sleep variable provided by the PSG, AWS, and FBC devices.

The Bland–Altman technique was used to plot the difference between FBC and AWS against the PSG measurement for every parameter, including TST, SOL, WASO, and SE [30]. The mean difference and 95% confidence interval are demonstrated [30]. A positive mean difference shows that a device overestimates the sleep variables when comparing them to the gold standard. A negative bias demonstrates that the variable is underestimated. A significant value indicates that the difference in scores between two devices changes according to the level of the measured sleep variable.

To quantify the agreement in sleep-wake categorization between AWS, FBC, and PSG, epoch-by-epoch(EBE) analyses were calculated for deriving three agreement measures: accuracy (ability to correctly classify epochs), sensitivity (ability to detect sleep), and specificity (ability to detect wakefulness) for each device setting (Table 1) [31]. Epoch-by-epoch comparisons were executed on pairwise matching. Sleep data for 8.1% (n = 3) of FBC were unavailable and were excluded from pairwise analyses. In order to assess potential biases in PSG, FBC, and

**Table 1. Calculation formula for sensitivity, specificity, and accuracy.**

| AWS/FBC | PSG | | |
| --- | --- | --- | --- |
| | Sleep | Wake | Total |
| Sleep | True Sleep(TS) | False Sleep(FS) | TS+FS |
| Wake | False Wake(FW) | True Wake(TW) | FW+TW |
| Total | TS+FW | FS+TW | TS+FS+TW+FW |

AWS data synchronization, we calculated EBE specificity by sliding the epoch alignment up to 90s (1–3 epochs forward and backward).

## Results

A total of 37 patients with chronic insomnia were chosen to participate in this study, including 20 women and 17 men (mean age = 48.8 ± 2.1). Their PSQI scores (mean score = 11.7 ± 2.8) and ISI scores (mean score = 19.8 ± 3.2) indicated mild to moderate sleep disturbance. The overall results, including sleep variables, are summarized in Table 2.

FBC: Fitbit Charge 4[TM]; AWS: Actiwatch Spectrum Pro; PSG: Polysomnography

### Bland–Altman mean difference analysis

The Bland–Altman technique was chosen to plot the difference between AWS and FBC against the gold standard PSG measurement for each sleep variable (Table 3). When the AWS values were compared to PSG, non-significant overestimations of TST (2.3, $p = 0.7549$), as well as non-significant underestimations of SE (-1.5%, $p = 0.2572$), SOL (-1.0, $p = 0.6907$), and WASO (-14.8, $p = 0.0509$), were identified. Corresponding Bland–Altman plots are shown below (Fig 1).

Direct comparison of FBC to AWS showcased remarkably lower assessment of SE (-3.6%, $p = 0.0008$) and TST (-13.8, $p = 0.0112$) for FBC, as well as comparably higher assessment of WASO (19.2, $p = 0.0001$). FBC and AWS demonstrated similar assessments of SOL (-0.8, $p = 0.7141$) (see Fig 3).

### Sensitivity, specificity, and accuracy

When compared epoch-by-epoch against PSG, AWS showed good sensitivity and accuracy, with poor specificity. Furthermore, FBC demonstrated a relatively better sensitivity and

**Table 2. Sleep variables of PSG, AWS, and FBC.**

| | PSG | AWS | FBC |
| --- | --- | --- | --- |
| **Sleep variables** | | | |
| TST(min) | 421.9±64.8 | 424.2±65.6 | 412.6±56.1 |
| SE(%) | 90.9±8.6 | 89.4±4.8 | 85.5±5.4 |
| WASO(min) | 42.7±44.5 | 27.8±17.3 | 47.6±23.5 |
| SOL(min) | 12.2±12.7 | 11.2±12.7 | 10.9±9.6 |
| LS(min) | 221.3±57.2 | - | 259.4±48.6 |
| DS(min) | 114.4±35.9 | - | 73.0±26.5 |
| REM(min) | 90.5±29.1 | - | 85.8±29.3 |

Sleep variables include: total sleep time (TST; min), sleep efficiency (SE; percent), sleep onset latency (SOL; min), wake after sleep onset (WASO; min), sleep stages N1 + N2 (LS; min), stage N3 (DS; min), and rapid eye movement sleep duration (REM; min)

**Table 3. Bias, SD, upper and lower agreement limits for Bland–Altman plots.**

| Variable | Device | Mean bias | Lower limit of agreement | Upper limit of agreement | p |
|---|---|---|---|---|---|
| TST(min) | AWS vs. PSG | 2.3 | -83.5 | 88.0 | 0.7549 |
| | FBC vs. PSG | -11.0 | -99.3 | 77.2 | 0.1620 |
| | AWS vs. FBC | -13.8 | -72.4 | 44.8 | 0.0112* |
| SE(%) | AWS vs. PSG | -1.5 | -16.6 | 13.7 | 0.2572 |
| | FBC vs. PSG | -4.9 | -21.2 | 11.4 | 0.0016* |
| | AWS vs. FBC | -3.6 | -14.6 | 7.5 | 0.0008* |
| SOL(min) | AWS vs. PSG | -1.0 | -30.3 | 28.3 | 0.6907 |
| | FBC vs. PSG | -1.8 | -22.0 | 18.4 | 0.2134 |
| | AWS vs. FBC | -0.8 | -25.4 | 23.8 | 0.7141 |
| WASO(min) | AWS vs. PSG | -14.8 | -102.4 | 72.7 | 0.0509 |
| | FBC vs. PSG | 2.8 | -66.6 | 72.3 | 0.6426 |
| | AWS vs. FBC | 19.2 | -31.4 | 69.7 | 0.0001* |
| LS(min) | FBC vs. PSG | 37.7 | -84.2 | 159.6 | 0.0012* |
| DS(min) | FBC vs. PSG | -41.4 | -122.6 | 39.8 | <0.0001* |
| REM(min) | FBC vs. PSG | -4.7 | -72.4 | 63.1 | 0.4371 |

Notes: *indicate statistically significant ($p < 0.05$).

When compared to PSG, FBC demonstrated a visible overestimation of LS (37.69, $p = 0.0012$), while significantly underestimating DS (-41.38, $p < 0.0001$) and SE (-4.9%, $p = 0.0016$). Non-significant overestimations of WASO (2.8, $p = 0.6426$), as well as non-significant underestimations of TST (-11.0, $p = 0.1620$), SOL (-1.8, $p = 0.2134$), and REM (-4.7, $p = 0.4371$), were also identified (see Fig 2).

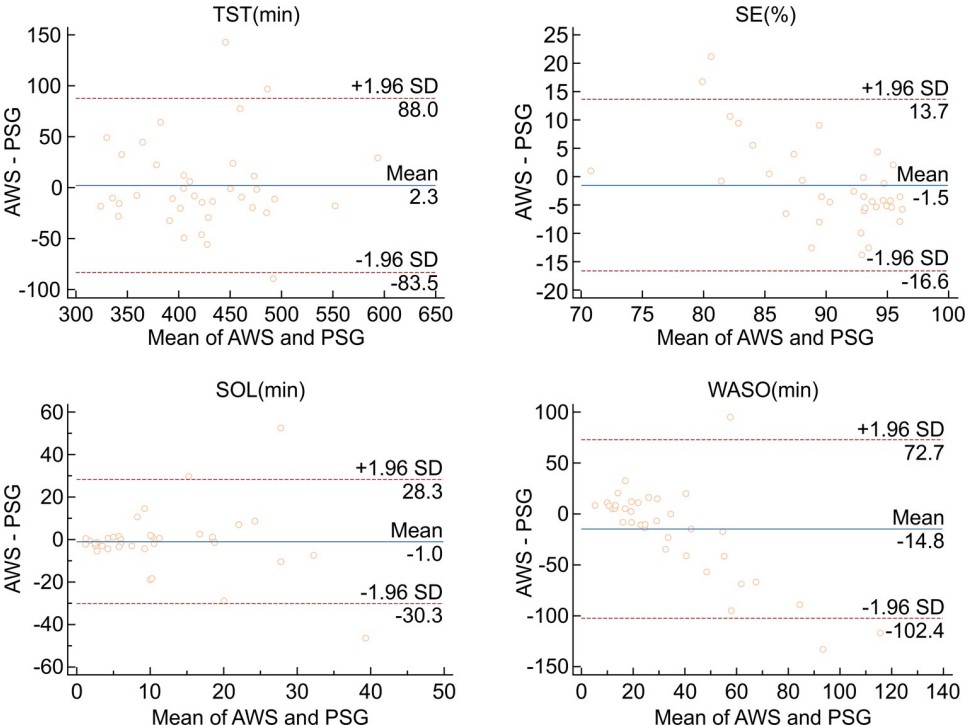

**Fig 1. Bland–Altman plot demonstrating mean bias, and upper and lower limits of agreement between PSG and AWS for all sleep variables (TST, SE, SOL, and WASO).**

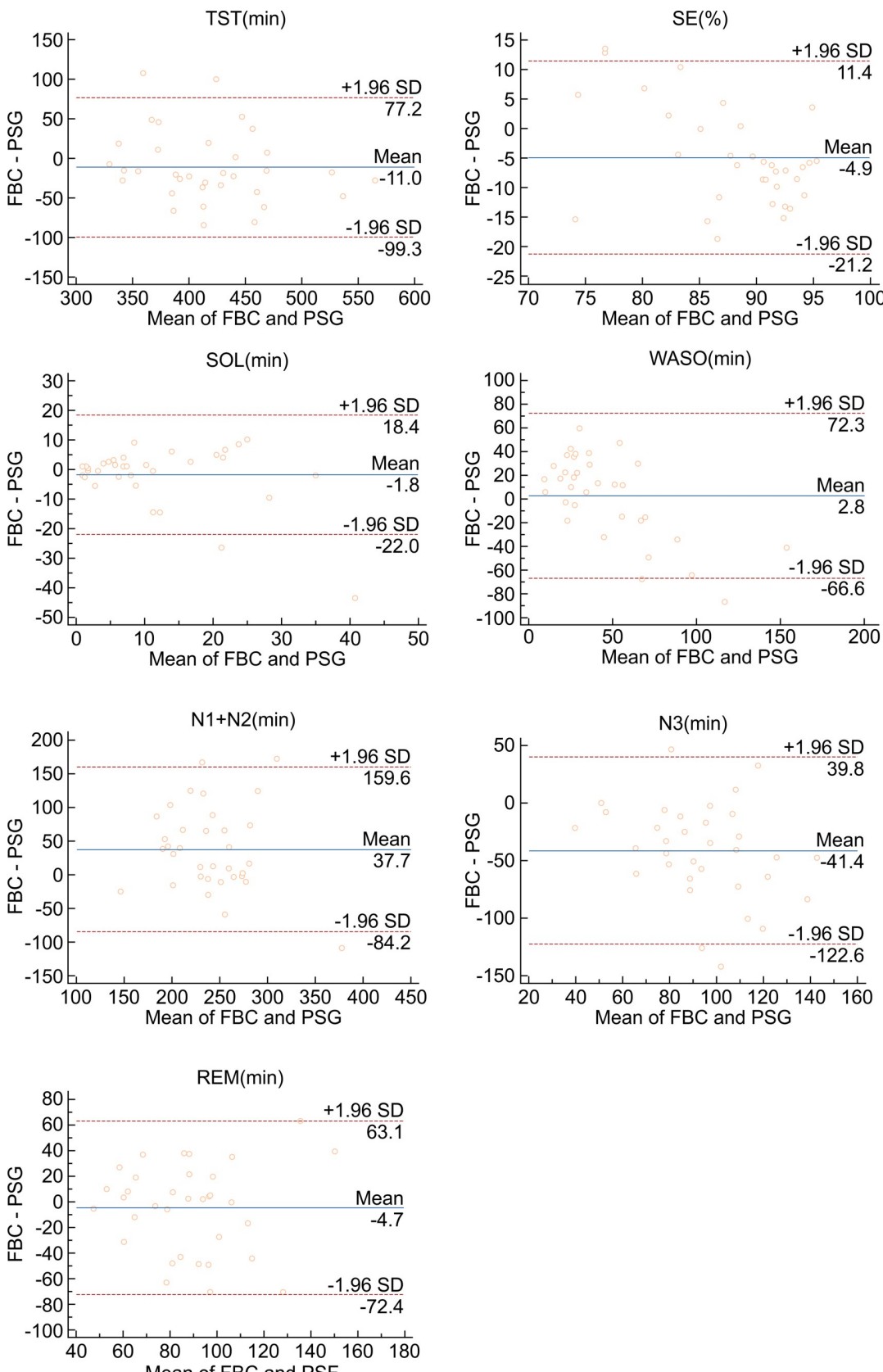

**Fig 2. Bland–Altman plot demonstrating mean bias, and upper and lower limits of agreement between PSG and FBC for all sleep variables (TST, SE, SOL, WASO, LS, DS, and REM).**

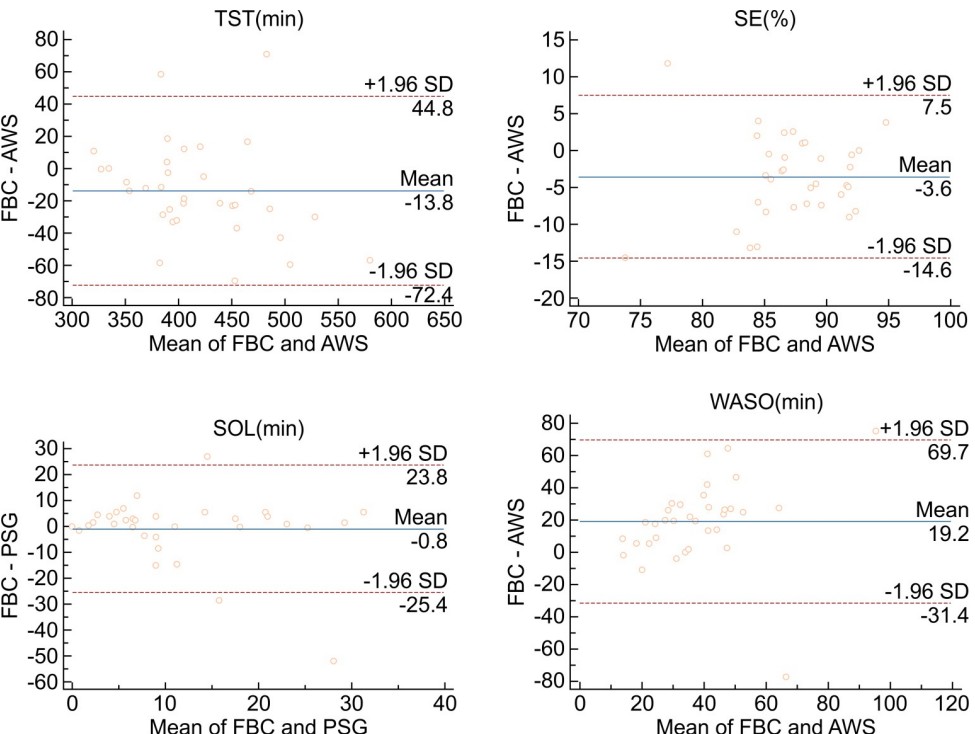

**Fig 3. Bland–Altman plot demonstrating mean bias, and upper and lower limits of agreement between AWS and FBC for all sleep variables (TST, SE, SOL, and WASO).**

accuracy, with moderate specificity. The results of the sensitivity, specificity, and accuracy are presented in Table 4.

## Discussion

It appears that our study is the first investigation to explore the accuracy of new generation fitness-trackers' (specifically FBC) use in Chinese patients with chronic insomnia. This disorder can remarkably influence both clinical care and research in the field of sleep medicine. In clinical practice, it is becoming increasingly popular to obtain sleep information from commercial activity trackers [18]. Thus, expounding the wearables' disadvantages and advantages is increasingly essential in the field of sleep disorder diagnosis and treatment. Additionally, clarifying these devices' potential to accurately estimate sleep is essential in interpreting longitudinal, field-based assessments of circadian rhythm in patients with sleep problems who may use these devices. Our research results show that FBC displays a similar performance in quantifying sleep variables and classifying sleep stages in chronic insomnia when compared to the gold standard of PSG. Although the outcomes are exciting, and indirectly enable a more accurate measuring of sleep in a larger sample size using consumer activity trackers, several limitations exist that need to be recognized.

**Table 4. Sensitivity, specificity, and accuracy in an epoch-by-epoch comparison of FBC and AWS with PSG.**

| Comparison | Sensitivity(%±SD) | Specificity(%±SD) | Accuracy(%±SD) |
|---|---|---|---|
| AWS vs. PSG | 92.6±15.7 | 35.7±20.1 | 86.9±10.1 |
| FBC vs. PSG | 89.9±4.0 | 62.2±26.2 | 86.5±5.4 |
| FBC vs. AWS | 89.1±4.8 | 75.7±23.3 | 87.9±5.3 |

## Consumer-grade Fitbit has caught up with research-grade actigraph

FBC is among the new generation of wearables that apply a multisensory technology to distinguish between sleep stages. In the current study, the primary outcomes demonstrate that FBC, in comparison to PSG, remarkably overestimates LS, while underestimating SE and DS. Moreover, FBC demonstrates no bias in the assessment of TST, WASO, SOL, or REM sleep relative to PSG and fully tracks nightly sleep cycles. However, the data about the accuracy of FBC in assessing sleep variables are conflicting. No significant differences were found between FBC and the gold standard PSG in assessing TST, WASO, and SOL. This is inconsistent with other studies' results [32, 33]. This may be due to the FBC high data loss as three participants' FBC data were lost. This corresponds to a recent longitudinal survey that showed high rates of data loss for the Fitbit [34]. Missing data were caused by patient factors (such as patients taking off the FBC during the night) or equipment failure (such as FBC recording ceased due to battery malfunction), which probably reduces the reliability of the FBC. However, when explaining the capability of FBC compared to different Fitbit models and other kinds of consumer activity trackers, we should not fully exclude certain influencing factors. For example, the participants included in the study or the type of study design and methodology could influence the consistency of results. Although no statistical difference was found, the absolute biases we observed for TST (approximately 11 min), SE (approximately 4.9%), and WASO (approximately 9 min) were in the lower tail of bias distributions shown by motion-based Fitbit devices (i.e., 7–67 min for TST, 2–15% for SE, and 6–44 min for WASO) [35]. These outcomes demonstrate that measuring TST and SE in normal populations with newer models of Fitbit devices tends to reduce the degree of overestimation. Our research results are consistent with the trend highlighted for consumer activity trackers' performance. This trend suggests increasing accuracy for the newer generation of consumer activity trackers [35, 36], possibly due to the implementation of a multi-sensor approach (integration of motion and photoplethysmographic data to quantify sleep and wake duration), and advancements in algorithm refinement.

Interestingly, the validated actigraph AWS performed comparably to PSG in assessing TST, SE, WASO, and SOL. However, AWS shows an incapability to exactly recognize wake epochs compared to PSG (specificity = 35.7%). The high reliability of AWS for estimating sleep variables corresponds to previous findings [37]. According to those studies, AWS is widely used both in research [38, 39] and clinical [40] settings. However, to study this reliability, users would have to remember bed and wake times and experienced researchers would be required to process the data from the software. This would be labor-consuming and the results would not be immediately obtainable. One of the disadvantages of actigraphy-based consumer activity trackers, for example AWS, is the relatively low specificity (accuracy to detect wakefulness). The specificity of FBC and AWS showed in this study differed significantly (62.2% and 35.7%). These findings also differed from previous reports that used the original Fitbit and Actiwatch and showed worse specificities for the former (19.8% and 38.9% respectively) [23]. In the current study, the specificity of FBC is 62.2%. Although there are no basic principles for determining a "good" or "bad" property [36], a specificity of 62.2% is better than that shown in previous studies measuring the accurateness of former Fitbit types. These studies depended only on movement to distinguish between sleep and wake (Fitbit "original", specificity of 20.0% [23]; Fitbit Flex, specificity of 35.0% and 36.0% [41]; Fitbit Ultra, specificity of 52.0% [18]; Fitbit Charge HR, specificity of 42.0% [42]; and Fitbit Charge 3™, specificity of 61.0% [43]). Previous studies have shown low specificity for former Fitbit types relative to PSG, usually less than 0.5 [44, 45]. However, the current study has found a specificity of 62.2%, which is within the specificity range (0.3 to 0.7), and is consistent with previous reports validating Fitbit devices against PSG [46]. To summarize, the results demonstrate that FBC's measurement of sleep

variables is comparable to that of PSG, as is the more traditional actigraphy. Moreover, these may also show more advantages in detecting wakefulness.

## Fitbit sleep staging

Several reports have discussed the accuracy of commercial fitness-trackers in measuring sleep stages against PSG. Our current findings demonstrate that FBC shows a significant overestimation of LS, while significantly underestimating DS. However, REM sleep estimation by FBC was accurate on average. Therefore, our findings replicate those of de Zambotti [43] and Menghini [25]. Owing to the development in transducer ability and signal processing technology, FBC has recently applied a multisensory information detection system for sleep detection. Such devices declare that sleep stages can be detected through various information sources besides motion, including HR and HRV. Theoretically, it is a reasonable assumption that the use of HR and HRV measurements are beneficial for sleep stage classification and the assessment of quiet wakefulness, when participants lay motionless on the bed. Particularly, sleep-stage specific shifting in autonomic activity, as monitored by HRV, is an accepted finding [47], with EEG and HRV measures tightly coupled overnight [48]. However, phasic sleep events (such as arousals and k-complexes), which are considered signs of sleep stage transitions in PSG [49], are accompanied by stereotypical HR fluctuations [50]. There is thus no reason to doubt the comparatively better specificity that the current research shows. It is a rising trend for superior accuracy in the new generation of multisensory activity trackers, owing to the use of other information sources besides movement, to demonstrate better ability to distinguish between sleep-wake states and sleep stages.

## Limitations

This investigation has certain limitations. As our study included participants who had been diagnosed with chronic insomnia for at least 3 months, the results may not be generalizable to participants with other sleep problems. Additionally, these conclusions may not be generalizable to longitudinal assessments of sleep-wake patterns using FBC since measurement was conducted for only one night in the current study. Considering that FBC would likely be most helpful in a longitudinal study, this study design should be used to assess multisensory wearables' accuracy over several nights in future studies. Moreover, depending on the participants included and the devices and algorithms used, variation in estimation of sleep parameters and staging among the studies would be great. Thus, conclusions from the current research may not be generalizable to other types of Actiwatches and Fitbits. A previous study has shown that various algorithms used to assess the same information generate a variety of results [16]. Thus, results from various studies assessing similar wearables with differing scoring systems would also not be generalizable.

## Conclusions

In conclusion, although our research shows that FBC cannot completely replace PSG in the quantification of sleep variables and the classification of sleep stage in patients with chronic insomnia, the user-friendly and low-cost wearables do show some comparable functions to PSG. The disadvantage in detecting wakefulness, complications surrounding the proprietary nature of the staging algorithm, privacy and regulatory concerns, and repeated information loss at the present time result in the restriction of clinical and research applications of the device in patients with chronic insomnia. Nevertheless, FBC has capacity as an estimator of sleep-wake patterns on par with standard actigraphy and PSG in a sleep laboratory. Additionally, FBC demonstrates better performance compared to other Fitbit models. This suggests

that, with future improvements, such multisensory activity trackers could be helpful when working with patients who have chronic insomnia.

## Acknowledgments

The authors thank Wang Pu, Zhang Lin, Xing Shasha, and Zhang Zhiqiang for their selfless assistance with the collecting, scoring, and staging of data. We express our gratitude to all the participants. We also thank Editage for polishing grammar.

## Author Contributions

**Data curation:** Sen Yang, Yuanli Guo, Peihua Lv.

**Formal analysis:** Yuanli Guo, Min Wang.

**Funding acquisition:** Yusheng Li.

**Methodology:** Xiaofang Dong, Sen Yang, Min Wang.

**Project administration:** Xiaofang Dong, Peihua Lv, Yusheng Li.

**Resources:** Yuanli Guo.

**Software:** Sen Yang, Peihua Lv.

**Supervision:** Yusheng Li.

**Validation:** Peihua Lv.

**Writing – original draft:** Xiaofang Dong.

**Writing – review & editing:** Min Wang, Yusheng Li.

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
