## [Decision Letter · Decision Letter 0]

25 May 2022

PONE-D-22-11757Validation of Fitbit Charge 4  for Assessing Sleep in Chinese Patients with Chronic Insomnia: A Comparison against Polysomnography and ActigraphyPLOS ONE

Dear Dr. Li,

Thank you for submitting your manuscript to PLOS ONE. After careful consideration, we feel that it has merit but does not fully meet PLOS ONE’s publication criteria as it currently stands. Therefore, we invite you to submit a revised version of the manuscript that addresses the points raised during the review process. Please ensure that your decision is justified on PLOS ONE’s publication criteria and not, for example, on novelty or perceived impact.

We look forward to receiving your revised manuscript.

Kind regards,

Christian Veauthier, M.D.

Academic Editor

PLOS ONE

Journal Requirements:

Reviewers' comments:

Reviewer's Responses to Questions

**Comments to the Author**

1. Is the manuscript technically sound, and do the data support the conclusions?

Reviewer #1: Yes

Reviewer #2: Yes

2. Has the statistical analysis been performed appropriately and rigorously? 

Reviewer #1: Yes

Reviewer #2: Yes

3. Have the authors made all data underlying the findings in their manuscript fully available?

Reviewer #1: No

Reviewer #2: Yes

4. Is the manuscript presented in an intelligible fashion and written in standard English?

Reviewer #1: Yes

Reviewer #2: Yes

5. Review Comments to the Author

Reviewer #1: The paper presents a validation study of a consumer sleep tracker, Fitbit Charge 4against two other methods. One method is the reference (gold) standard polysomnography, and the other is an often used medical device, actigraphy. The validation was performed in 37 patients suffering from insomnia. This is an important research question because more and more patients come to sleep centers and show their Fitbit results to a sleep physician in order to ask for a polysomnography.

In the abstract and in the text, there is an error: SOL is not sleep of latency but sleep onset latency.

Paragraph 2.3: For the sleep recording a few more technical details are missing. You say, that a computerized sleep recorder by Embla was used. However, there are many different versions marketed. Please specify exactly which type (name) of hardware and which software (Somnologica, Remlogic) including version number had been used here. If different sleep recorders had been used, please mention this.

I am happy that you specified the version number for the Actiware software. The same is required for the other software involved.

Paragraph 2.4: For the sleep scoring, you say a trained neurophysiologist scored the PSG records according to AASM. However, there are several versions of AASM with important differences. Please specify exactly which AASM version was used.

Paragraph 2.4: Regarding the visual scoring by the trained neurophysiologist, I like to know the inter- and intrascorer reliability of the person. This is, because we know that there is always a limited accuracy in the visual scoring.

Because the subjects of your study suffered from insomnia, I would like to know their ISI (Insomnia severity index) score. Did you assess this important value?

How many nights did you perform polysomnography? Usually the first night is used for adaption and the second night is used for evaluation.

How long did the patients wear the Actiwatch? Usually this is for two weeks.

How long did the patients wear the Fitbit. Often this is for months.

How did you take care of synchronization of the data between the three systems? I would like to know how accurate was the synchronization of a specific epoch? Each epoch is 30 seconds and a synchronization error of perhaps 5 or 10 seconds could give different scoring results.

Was the Fitbit and the Actiwatch mounted on the same arm or different arms? This might result in some differences?

The number of patients is relatively small. Why did you investigate 37 patients only? There are publications which require that validation studies for devices recording sleep should have at least 64 patients.

Presentation of the results is good. Especially the Bland-Altman plots show the limited value of the indirect devices. I do find the AWS against FBC plots most interesting. Table 3 is a good summary of comparisons.

The discussion is well to the different points. I would recommend to structure it more to the different comparisons and also to show limitations better.

Reviewer #2: This is a well-written paper, which addresses an important issue, the measurement of sleep with new technologies.

I have the following comments:

The authors should think about presenting Bland-Altmann-Plots, sensitivity, specificity, and accuracy data just for the comparison of the Fitbit with PSG as well as for the AWS with PSG. The reason for omitting the comparison Fitbit with AWS is that none of these devices could be assumed to be the “reference” of sleep measurement.

On the other hand, it would be interesting to have a 4x4 table with the total numbers (or percentage) of epochs assigned to stages light sleep, deep sleep, dream sleep, and Wake illustrating the agreement of Fitbit with PSG for correct classification of sleep structure.

Of course, wearables like the new Fitbit device have a great potential in assessing sleep in big populations. Although the authors should also discuss some limitations of those devices:

As they are no medical devices and most of them do not tend to become one manufacturers typically do not specify sensors, the algorithms used for data processing, statistical analysis, and reporting. This limit the interpretability and validity of results. Examples are the facts that authors had to collect sleep stages manually from summary figures and that without reason 8.1% of Fitbit recordings were unavailable for analysis.

Another problem with consumer devices is that if one device was tested in a study upon publication of results it might be replaced on the market by a new one with different hardware as well as software already. An example the authors stated is the Fitbit device with five (!) previous versions. A successor - the Fitbit Charge5 - is on the market already …

In summary this make it impossible to perform large studies in different cohorts or even to perform replication studies.

Minor:

Results, 3.3.

- prevent redundancy of data presentation (in text as well as in table 4)

Table 2

- I suggest to either put demographics into extra table or in text

6. PLOS authors have the option to publish the peer review history of their article (what does this mean?). If published, this will include your full peer review and any attached files.

Reviewer #1: **Yes: **Thomas Penzel

Reviewer #2: No

---

## [Decision Letter · Decision Letter 1]

13 Sep 2022

Validation of Fitbit Charge 4  for Assessing Sleep in Chinese Patients with Chronic Insomnia: A Comparison against Polysomnography and Actigraphy

PONE-D-22-11757R1

Dear Dr. Li,

We’re pleased to inform you that your manuscript has been judged scientifically suitable for publication and will be formally accepted for publication once it meets all outstanding technical requirements.

Kind regards,

Christian Veauthier, M.D.

Academic Editor

PLOS ONE

Additional Editor Comments (optional):

Reviewers' comments:

Reviewer's Responses to Questions

**Comments to the Author**

1. If the authors have adequately addressed your comments raised in a previous round of review and you feel that this manuscript is now acceptable for publication, you may indicate that here to bypass the “Comments to the Author” section, enter your conflict of interest statement in the “Confidential to Editor” section, and submit your "Accept" recommendation.

Reviewer #1: All comments have been addressed

Reviewer #2: All comments have been addressed

2. Is the manuscript technically sound, and do the data support the conclusions?

Reviewer #1: Yes

Reviewer #2: Yes

3. Has the statistical analysis been performed appropriately and rigorously? 

Reviewer #1: Yes

Reviewer #2: Yes

4. Have the authors made all data underlying the findings in their manuscript fully available?

Reviewer #1: Yes

Reviewer #2: Yes

5. Is the manuscript presented in an intelligible fashion and written in standard English?

Reviewer #1: Yes

Reviewer #2: Yes

6. Review Comments to the Author

Reviewer #1: Thank you for considering the comments by the reviewers. All questions and concerns were addressed. The revision did help to improve the manuscript.

Reviewer #2: All of my comments were answered with satisfaction.

Minor: Line 170: Since you stated that one person scored the data I suggest to write "reviewed by a trained technician" instead of "reviewed by trained by technicians"

7. PLOS authors have the option to publish the peer review history of their article (what does this mean?). If published, this will include your full peer review and any attached files.

Reviewer #1: No

Reviewer #2: No

---

## [Editor Report · Acceptance letter]

5 Oct 2022

PONE-D-22-11757R1 

Validation of Fitbit Charge 4 for assessing sleep in Chinese patients with chronic insomnia: A comparison against polysomnography and actigraphy 

Dear Dr. Li:

I'm pleased to inform you that your manuscript has been deemed suitable for publication in PLOS ONE. Congratulations! Your manuscript is now with our production department. 

Kind regards, 

on behalf of

Dr. Christian Veauthier 

Academic Editor

PLOS ONE